# Logit Margin Matters: Improving Transferable Targeted Adversarial Attack by Logit Calibration

## Abstract

Previous works have extensively studied the transferability of adversarial samples in untargeted black-box scenarios. However, it still remains challenging to craft the targeted adversarial examples with higher transferability than non-targeted ones. Recent studies reveal that the traditional Cross-Entropy (CE) loss function is insufficient to learn transferable targeted perturbations due to the issue of vanishing gradient. In this work, we provide a comprehensive investigation of the CE function and find that the logit margin between the targeted and non-targeted classes will quickly obtain saturated in CE, which largely limits the transferability. Therefore, in this paper, we devote to the goal of enlarging logit margins and propose two simple and effective logit calibration methods, which are achieved by downscale the logits with a temperature factor and an adaptive margin, respectively. Both of them can effectively encourage the optimization to produce larger logit margins and lead to higher transferability. Besides, we show that minimizing the cosine distance between the adversarial examples and the targeted classifier can further improve the transferability, which is benefited from downscale logits via L2-normalization. Experiments conducted on the ImageNet dataset validate the effectiveness of the proposed methods, which outperforms the state-of-the-art methods in black-box targeted attacks. The source code of our method is available at Link.

## 1 Introduction

In the past decade, deep neural networks (DNNs) have achieved remarkable success in various fields, *e.g.*, image classification [24], image segmentation [19], and object detection [23]. However, Goodfellow *et al.* [5] reveal that the DNNs are vulnerable to adversarial attacks, in which adding imperceptible disturbances to the input can lead the DNNs to make an incorrect prediction. Many following approaches [3, 4, 1, 27, 29] have been proposed to construct more destructive adversarial samples for investigating the vulnerability of the DNNs. [5, 18] also show that the adversarial samples are transferable across different networks, raising a more critical robustness threat under the black-box scenarios. Therefore, it is vital to explore the vulnerability of the DNNs, which is very useful for designing robust DNNs.

Currently, most of the works [3, 29, 16, 10, 28, 6] have been devoted to the untargeted black-box attack, in which adversarial examples are crafted to fool unknown CNN models to predict unspecified incorrect labels. For example, [3, 29] leveraged input-level transformation or augmentation to improve the non-targeted transferability. [10] proposed a powerful intermediate feature-level attack. [28, 6] demonstrated that backpropagating more gradients through the skip-connections can increase the transferability. Despite the success in non-targeted cases, the targeted transferability remains challenging, which requires eliciting the black-box models into a pre-defined target category label.

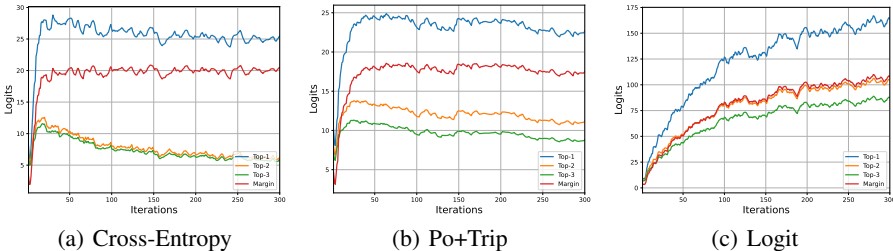

|  |  |  |
|---|---|---|
| (a) Cross-Entropy | (b) Po+Trip | (c) Logit |

Figure 1: The average Top-3 logits and logit margin of 50 adversarial samples trained by the Cross-Entropy, Po+Trip and Logit loss functions for crafting the ResNet-50. (* Training and computation details of this figure are in Section 3.1)

For learning the transferable adversarial samples in untargeted cases, most methods have leveraged the Cross-Entropy (CE) as the loss function. However, [15, 30] recently showed that the CE loss is insufficient for learning the adversarial perturbation in the targeted case due to the issue of vanishing gradient. To deal with this issue, [15] adopt the Poincaré distance to increase the gradient magnitude during the optimization adaptively. [30] demonstrated that an effortless logit loss equal to the negative value of the targeted logits could alleviate the gradient issue and achieve surprisingly strong targeted transferability. Besides, [30] also showed that optimizing with more iterations can significantly increase the targeted transferability. Although [30] demonstrated that continually enlarging the logits of the targeted class can improve the transferability of adversarial samples, it still does not thoroughly analyze the insufficient issue in the CE loss function.

In this study, we take a closer look at the vanishing gradient issue in the CE and find that the logit margin between the targeted and non-targeted classes will quickly get saturated during the optimization (as shown in Fig. 1(a)). Moreover, this issue will influence the performance of the perturbations and thus essentially limit the transferability. Specifically, along with the training iterations in CE, we observe that the logits of the targeted and non-targeted classes increase rapidly in the first few iterations. However, after reaching the peak, the logit margin between the targeted and non-targeted classes will get saturated, and further training will decrease the logits simultaneously to maintain this margin. This phenomenon is mainly due to the fact that the softmax function in CE will approximately output the probability of the target class to 1 when reaching the saturated margin (*e.g.*, 10). Thus, it raises the problem that the transferability will not be further increased even optimized with more iterations. While in practice, we are encouraged to increase the transferability by maximizing both the logit for the targeted class and its margin against other non-targeted classes to cross the decision boundaries of other black-box models.

In this paper, we devote to enlarging logit margins to alleviate the above saturation issue in CE. Inspired by the temperature-scaling used in the knowledge distillation [8], a higher temperature $T$ will produce a softer probability distribution over different classes. We firstly leverage this scaling technique into the targeted adversarial attack to calibrate the logits. Then the logits margin between the targeted and non-targeted classes will not be saturated after only a few iterations and will keep improving the transferability. On the other aspect, instead of using a constant $T$, we further explored an adaptive margin-based calibration by scaling the logits based on the logit margin of the target class and the highest non-target class. In addition, we also investigate the effectiveness of calibrating the targeted logit into the unit length feature space by L2-normalization, which is equivalent to minimizing the angle between the adversarial examples and the targeted classifier.

Finally, we conduct experiments on the ImageNet dataset to validate the effectiveness of the logits calibration for crafting transferable targeted adversarial examples. Experimental results demonstrated that the calibration of the logits helps achieve a higher attack success rate than other state-of-the-art methods. Additionally, we tested the logit calibration in Generative Adversarial Networks (GANs)-based TTP method [21] to train the target-class-specific generators, which is also beneficial for increasing the transferability in the resource-intensive method.

## 2 Related Works

In this section, we give a brief introduction of the related works from the following two aspects: *untargeted black-box attacks* and *targeted attacks*.

## 2.1 Untargetd Black-box Attacks

After [25] exposed the vulnerability of deep neural networks, many attack methods [29, 4] have been proposed to craft highly transferable adversaries in the non-targeted scenario. We first review several gradient-based attack methods that focus on enhancing the transferability against black-box models.

**Iterative-Fast Gradient Sign Method (I-FGSM)** [14] is an iterative version of FGSM [5], which adds a small perturbation with a small step size $\alpha$ in the gradient direction iteratively:

$$\hat{x}_0 = x, \quad \hat{x}_{i+1} = \hat{x}'_i + \alpha \cdot \text{sign}(\nabla_{\hat{x}} J(\hat{x}'_i, y)), \tag{1}$$

where $\hat{x}'_i$ denotes the adversarial image in the $i_{th}$ iteration, $\alpha = \epsilon/T$ ensures the adversaries be constrained within an upper-bound perturbation $\epsilon$ through the $l_p$-norm when optimized by $T$ iterations.

Following the seminal I-FGSM [14], a series of methods have been proposed to improve the transferability of attacking black-box models from different aspects, *e.g.*, gradient-based, input augmentation-based. For example, the **Momentum Iterative-FGSM (MI-FGSM)** [3] introduces a momentum term to compute the gradient of the I-FGSM, encouraging the perturbation is updated in a stable direction. The **Translation Invariant-FGSM (TI-FGSM)** [4] adopts a predefined kernel $W$ to convolve the gradient $\nabla_{\hat{x}} J(\hat{x}'_i, y)$ at each iteration $t$, which can approximated the average gradient over multiple randomly translated images of the input $\hat{x}_t$. On the other aspects, the **Diverse Input-FGSM (DI-FGSM)** leveraged the random resizing and padding to augmentation the input $\hat{x}_t$ at each iteration. Currently, most targeted attack methods [15, 30, 21] simultaneously use the MI, TI and DI to form a strong baseline with better transferability.

## 2.2 Targeted Attacks

Targeted attacks are different from non-targeted attacks, which need to change the decision to a specific target class. [13] integrates the above non-targeted attack methods into targeted attacks to craft targeted adversarial examples. However, the performance is limited because it is insufficient to fool the black-box model only by maximizing the probability of the target class with the CE loss.

**Po+Trip** [15] found the insufficiency is mainly due to vanishing gradient issue in CE. Then, [15] leverage dthe Poincaré space as the metric space and further utilized Triplet loss to improve targeted transferability by forcing adversarial example toward the target label and away from the ground-truth label. To further address this gradient issue, **Logits** [30] adopts a simple and straightforward idea by directly maximizing the target logit to pull the adversarial examples close to the target class, which can be expressed as:

$$L_{Logit} = -z_t(\boldsymbol{x}'), \tag{2}$$

where $z_t(\cdot)$ is the output logits of the target class.

On the other hand, many studies employ resource-intensive approaches to achieve targeted attack, which train target class-specific models (auxiliary classifiers or generative models) on additional large-scale data. For example, the FDA methods [12, 11] used the intermediate feature distributions of CNNs to boost the targeted transferability by training class-specific auxiliary classifiers to model layer-wise feature distributions. The GAP [22] trained a generative model for crafting targeted adversarial examples. Subsequently, [20] adopted a relativistic training objective to train the generative model for improving attack performance and cross-domain transferability. Recently, the TTP [21] utilized the global and local distribution matching for training target class-specific generators for obtaining high targeted transferability. However, the TTP requires actual data samples from the target class and brings expensive training costs. Different from the above methods, we introduce three simple and effective logit calibration methods into the CE loss function, which can achieve competitive performance without additional data and training.

## 3 Method

**Problem Definition** Given a white-box surrogate model $\mathbb{F}_s$ and an input $x$ not from the targeted class $t$, our primary goal is to learn an imperceptible perturbation $\delta$ that can fool the $\mathbb{F}_s$ to output the target $t$ for $\hat{x} = x + \delta$. Besides, the prediction of $\hat{x}$ will also be $t$ when feeding to other unknown black-box surrogate models. The $l_\infty$-norm is usually used to constrained the $\delta$ within an upper-bound $\epsilon$, denoted as $||\delta||_\infty \leq \epsilon$.

For the surrogate model $\mathbb{F}_s$, we denoted the feature for the final classification layer of the input $x$ as $\phi(x)$. The logit $z_i$ of a category $i$ is computed by $z_i = W_i^T \phi(x) + b_i$, the $W_i$ and $b_i$ are the classifier weights and bias. The corresponding probability $p_i$ after the softmax is $p_i = \frac{e^{z_i}}{\sum e^{z_j}}$.

## 3.1 Logit Margin

When successfully attacked the $\mathbb{F}_s$, the logit $z_t$ of the target class will be higher than the logits $z_{nt}$ of any other non-target class in the classification task. Their logit margins can be computed by,

$$G(\phi(\hat{x})) = z_t - z_{nt} = W_t^T \phi(\hat{x}) + b_t - W_{nt}^T \phi(\hat{x}) + b_{nt}. \tag{3}$$

[15, 30] showed that it is insufficient to obtain transferable targeted adversarial samples that are only close to the target class while not away from true class and other non-targeted classes. Based on this property, it encourages us to continually enlarge this logit margin to increase the separation between the targeted and other non-targeted classes.

To have a better understanding of the relationship between the logit margins and the targeted transferability, we visualize the average Top-3 logits (1 targeted class and other two non-targeted classes) of 50 random adversarial samples trained for crafted ResNet50 by the CE, Po+Trip, and the Logit loss functions with MI, DI and TI following [30]. We also compute the average logit margin of the targeted class against the Top-20 non-targeted classes. The logit and margin are shown in Figure 1, and the transferability from ResNet50 to VGG16 is plot in Figure 2.

From Figure 1, we can observe that the logits of the targeted class and the Top-2 non-targeted classes increase rapidly in the first few iterations for the CE and Po+Trip loss, as well as their logit margins. When reaching the peak, the margin is saturated, and the logits start to decrease simultaneously to maintain the saturated margin. By comparing the CE and Po+Trip, the Po+Trip needs fewer more iterations to reach the saturated status and thus shows a marginal better transferability than CE, as shown in Figure 2. In comparison, the Logit loss function will keep increasing the logits of the targeted category and the logit margin. Thus, the Logit loss function shows a much better targeted-attack success rate than CE and Po+Trip. On the other hand, the Logit loss also significantly increases the logits for other non-targeted classes.

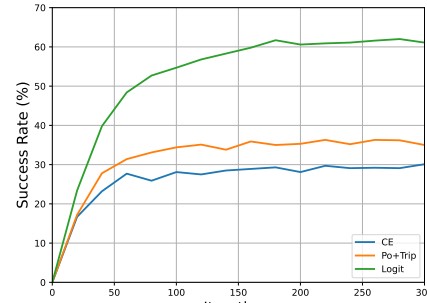

Figure 2: The targeted attack success rate (%) on VGG-16 by using the ResNet-50 as the surrogate model.

To further analyze why the CE loss function saturated to this logit margin and explore the effectiveness of increasing the margin during training, in the following sections, we will revisit the cross-entropy loss function and introduce the logit calibration to achieve this goal.

## 3.2 Revisiting the Cross-Entropy Loss

Firstly, our objective is to maximize the logit margin in Eq. 3. After computing the gradient w.r.t. to $\phi(\hat{x})$, we can get

$$\frac{\partial G}{\partial \phi(x)} = W_t - W_{nt}. \tag{4}$$

This gradient indicates that the adversarial feature $\phi(\hat{x})$ needs to move towards the target class while apart from those non-target classes. Next, we compute the gradient w.r.t. to $\phi(\hat{x})$ in the Cross-Entropy loss function

$$L_{CE} = -\log(p_t) = -z_t + \log(\sum e^{z_k}), \tag{5}$$

and get the gradient $\frac{\partial L_{ce}}{\partial \phi(\hat{x})}$ as

$$\frac{\partial L_{ce}}{\partial \phi(\hat{x})} = -\frac{\partial z_t}{\partial \phi(\hat{x})} + \frac{1}{\sum e^{z_k}} \cdot \frac{\partial \sum e^{z_k}}{\partial \phi(\hat{x})} \tag{6}$$

$$= -\frac{\sum e^{z_i}}{\sum e^{z_k}} \cdot \frac{\partial z_t}{\partial \phi(\hat{x})} + \frac{1}{\sum e^{z_k}} \sum e^{z_i} \frac{\partial z_i}{\partial \phi(\hat{x})}$$

$$= \sum \frac{e^{z_i}}{\sum e^{z_k}} \cdot \left(\frac{\partial z_i}{\partial \phi(\hat{x})} - \frac{\partial z_t}{\partial \phi(\hat{x})}\right) = \sum -p_i(W_t - W_i).$$

From Eq. 6, we actually can find the CE loss function is designed to adaptively optimize the $\phi(\hat{x})$ towards $W_t$ and away from other $W_i$. However, after being optimized for several iterations, the $p_i$ of the non-targeted class will quickly approximate to 0 and then significantly vanish the $W_t - W_i$.

Let's consider the case only with 2 classes ($t$ and $nt$), we have the probabilities $p_t$ and $p_{nt}$ as:

$$p_t = \frac{e^{z_t}}{e^{z_t} + e^{z_{nt}}} = \frac{1}{1 + e^{-(z_t - z_{nt})}}, \tag{7}$$

$$p_{nt} = \frac{e^{z_{nt}}}{e^{z_t} + e^{z_{nt}}} = \frac{1}{1 + e^{(z_t - z_{nt})}}. \tag{8}$$

As shown in Figure 3, the $p_t$ will get close to 1 when $z_t - z_{nt} > 6$ (*e.g.*, $p_{nt} \approx 2e^{-9}$ when $z_t - z_{nt} = 20$). In such a context, the gradient will significantly vanish. Recall that, in the CE loss function (Figure 1 (a)), the logit margin between Top-1 and Top-2 logits first increases rapidly but will reach saturated status when approaching a certain value. This further indicates that the optimization of the CE loss function is largely restrained when the logit margin reaches a certain value.

To this end, we raise the question *if we explicitly enforce the optimization to enlarge the logit margin ($z_t - z_{nt}$), could we get better transferable targeted adversarial samples?*

To answer this, we propose to downscale the $z_t - z_{nt}$ by a factor $s$ in the CE and extent the informative optimization for more iterations. Since in such circumstance, $z_t - z_{nt}$ will be enlarger by the factor $s$. Specifically, suppose that the optimization will be saturated when $z_t - z_{nt}$ reaches a certain value $v$. Using $z_t - z_{nt}$ and $\frac{z_t - z_{nt}}{s}$ in the CE will both approach the saturated value of $v$. Then, it is easy to infer that, for the latter case, $z_t - z_{nt}$ will be $v \times s$.

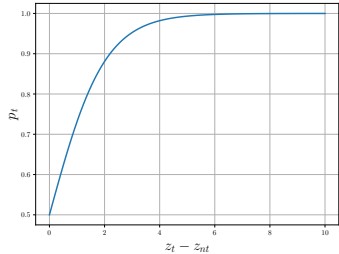

Figure 3: The probability of $p_t$ under different $z_t - z_{nt}$.

## 3.3 Calibrating the Logits

To downscale the $z_t - z_{nt}$ during the optimization, we investigate three different types of logit calibrations in this study, *i.e.*, Temperature-based, Margin-based, and Angle-based.

### 3.3.1 Temperature-based

Inspired by the Temperature-scaling used in the Knowledge distillation [8], our first logit calibration directly downscale the logits by a constant temperature factor $T$,

$$\tilde{z}_i = \frac{z_i}{T}. \tag{9}$$

After introducing the $T$, the probability distribution $\boldsymbol{p}$ will be more softer over different classes. The corresponding gradient can be compute by:

$$\frac{\partial L_{ce}^T}{\partial \phi(\hat{x})} = \frac{e^{z_j/T}}{\sum e^{z_j/T}} \cdot \frac{1}{T}\left(\frac{\partial z_j}{\partial x} - \frac{\partial z_t}{\partial \hat{x}}\right) = \sum -\hat{p}_i \frac{(W_t - W_i)}{T}. \tag{10}$$

The $\hat{p}_i$ will not quickly approach to 0 after only a few iterations.

In Figure 4 (a)(b), we visualized the logits of using $T = 5$ and $T = 20$. We can find that targeted logits and the logit margin will keep increasing as the same as the Logit in Figure 1. Meanwhile, the trend of $T = 20$ is very similar with the Logit [30] and we show that the Logit loss function can be considered as a special case of calibrating the logits with a large $T$ in the supplementary.

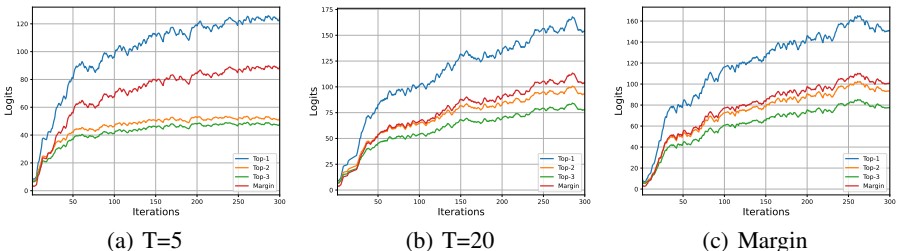

|     |     |     |
| :-: | :-: | :-: |
| (a) T=5 | (b) T=20 | (c) Margin |

Figure 4: The average Top-3 logits and logit margin of 50 adversarial samples after the logit calibration for crafting the ResNet-50.

#### 3.3.2 Margin-based

The previous Temperature-based logit calibration contains a hype-parameter $T$, which could be different for different surrogate model $\mathbb{F}_s$. To migrate this issue, we further introduce an adaptive margin-based logit calibration. Specifically, we calibrate the logits by using the margin between the Top-2 logits in each iteration, denoted as:

$$\tilde{z}_i = \frac{z_i}{\hat{z}_1 - \hat{z}_2},\tag{11}$$

where $\hat{z}_1$ and $\hat{z}_2$ are the Top-1 and the Top-2 logit, respectively.

In this Margin-based logit calibration, we will enforce the $p_t$ and $p_{\hat{1}}$ of the Top-1 non-target class at each iteration meet the following constraints:

$$p_t = \frac{1}{1 + \sum_{i \neq t} e^{-(\tilde{z}_t - \tilde{z}_i)}} < \frac{1}{1 + e^{-1}},\tag{12}$$

$$p_{\hat{1}} = \frac{1}{e^{\tilde{z}_{\hat{1}} - \tilde{z}_t} + \sum_{i \neq t} e^{\tilde{z}_i - \tilde{z}_{\hat{1}}}} > \frac{1}{N - 1}(1 - \frac{1}{1 + e^{-1}}).\tag{13}$$

Then, it can adaptively deal with the vanishing gradient issue in the original CE loss function. The logits and the margin is shown in Figure 4 (c).

#### 3.3.3 Angle-based

On the other aspect, different $W_t$ usually has a different norm. To further alleviate the influence of various norms, we calibrate the logit into the feature space with unit length by L2-normalization, $\frac{W_i^T \phi(\hat{x}) + b_i}{||W_i|| ||\phi(x)||}$. If omit the $b_i$, this calibration is compute the $cos(\theta)$ between $\phi(\hat{x})$ and $W_i$, and we term it as angle-based calibration. *Since, this angle-based calibration will bound each logit smaller than one.* Instead of using the CE loss function, we directly minimize the angle between the $\phi(\hat{x})$ and the targeted $W_t$. The optimization loss function is:

$$L_{cosine} = -\frac{W_t^T \phi(\hat{x})}{||W_t|| ||\phi(\hat{x})||}.\tag{14}$$

The angle-based classifiers have been widely using in Face-Recognition task [17, 2]. In the experiments, we evaluate the performance of using different logit calibrations and their mutual benefits.

### 4 Experiments

**Experimental Setup** In this section, we evaluate the effectiveness of logit calibration for improving transferable targeted adversarial attack. Following the recent study [30], we conduct the experiments on the difficult ImageNet-Compatible Dataset[1]. This dataset contains 1,000 images with 1,000 unique class labels corresponded to the ImageNet dataset. We implement our methods based on the source

---

[1] https://github.com/cleverhans-lab/cleverhans/tree/master/cleverhans_v3.1.0/
examples/nips17_adversarial_competition/dataset

Table 1: The targeted transfer success rates (%) in the single-model transfer scenario. (Results with 20/100/300 iterations are reported, and the highest one at 300 iterations is showed **bold**.)

| Attack | Surrogate Model: ResNet50 | | | Surrogate Model: Dense121 | | |
| --- | --- | --- | --- | --- | --- | --- |
| | →Dense121 | →VGG16 | →Inc-v3 | →Res50 | →VGG16 | →Inc-v3 |
| CE | 27.0/40.2/42.7 | 17.4/27.6/29.1 | 2.3/4.1/4.6 | 12.3/17.2/18.4 | 8.6/10.5/10.9 | 1.6/2.3/2.8 |
| Po+Trip | 27.9/51.2/54.8 | 17.9/35.5/34.7 | 3.2/6.8/7.8 | 11.0/14.8/15.0 | 7.3/9.2/8.6 | 1.6/2.8/2.8 |
| Logit | 31.4/64.0/71.8 | 23.8/55.0/62.4 | 3.1/8.6/10.9 | 17.4/38.6/43.5 | 13.7/33.8/37.8 | 2.3/6.6/7.5 |
| T=5 | 33.3/69.9/**77.8** | 24.8/59.9/66.1 | 3.1/9.4/**12.2** | 19.3/43.4/47.5 | 14.6/36.6/39.4 | 2.3/7.3/8.8 |
| T= 10 | 31.6/68.5/77.0 | 23.6/58.5/**66.4** | 2.8/9.4/11.6 | 17.9/43.2/**49.3** | 13.4/36.8/**41.5** | 2.2/7.7/8.8 |
| Margin | 33.3/65.8/76.5 | 23.1/58.6/65.7 | 3.0/9.5/12.2 | 18.8/42.8/47.2 | 14.5/36.5/41.4 | 2.5/7.7/**9.4** |
| Angle | 38.9/72.5/77.2 | 29.2/60.7/65.2 | 4.4/10.7/11.1 | 20.6/43.2/47.8 | 16.5/35.7/39.3 | 3.0/7.7/8.9 |

| Attack | Surrogate Model: VGG16 | | | Surrogate Model: Inc-v3 | | |
| --- | --- | --- | --- | --- | --- | --- |
| | →Res50 | →Dense121 | →Inc-v3 | →Res50 | →Dense121 | →VGG16 |
| CE | 0.5/0.3/0.6 | 0.6/0.3/0.3 | 0/0/0.1 | 0.7/1.2/1.8 | 0.6/1.3/1.9 | 0.4/0.8/1.3 |
| Po+Trip | 0.7/0.6/0.7 | 0.7/0.6/0.5 | 0.1/0.1/0.1 | 1.0/1.6/1.7 | 0.6/1.7/2.5 | 0.7/1.2/1.8 |
| Logit | 3.4/9.9/11.6 | 3.5/12.0/13.9 | 0.3/1.0/**1.3** | 0.6/1.1/2.0 | 0.6/1.9/3.0 | 0.6/1.5/2.8 |
| T=5 | 3.1/7.0/6.9 | 3.3/7.6/7.8 | 0.2/0.9/0.8 | 0.7/1.7/2.1 | 0.5/1.9/**3.3** | 0.4/1.6/2.6 |
| T= 10 | 3.6/9.0/9.7 | 3.4/10.5/11.7 | 3.2/1.1/1.3 | 0.5/1.3/1.9 | 0.6/2.0/2.7 | 0.4/1.5/**2.8** |
| Margin | 3.3/10.3/**12.0** | 3.5/12.5/**14.5** | 0.3/1.1/**1.3** | 0.5/1.4/1.7 | 0.7/2.1/3.1 | 0.5/1.7/2.7 |
| Angle | 0.4/0.7/0.5 | 0.6/0.4/0.5 | 0/0/0.1 | 0.8/1.8/**2.6** | 0.8/2.2/3.0 | 0.9/1.7/2.4 |

code[2] provided by the Logit [30]. The same four diverse CNN models are used for evaluation, *i.e.*, ResNet-50 [7], DenseNet-121 [9], VGG-16 with Batch Normalization [24] and Inception-v3 [26]. The perturbation is bounded by $L_\infty \leq 16$. The TI [4], MI [3] and DI [29] were used for all attacks, and $||W||_1 = 5$ is set for TI. The I-FSGM is adopted for optimization with the $\alpha = 2$. The attacks are trained for 300 iterations on a NVIDIA-2080 Ti GPU. We run all the experiments for 5 times, and report the average targeted transfer success rates (%). More experimental results can be found in the supplementary.

## 4.1 Comparison with Other Methods in Single-Model Transfer

We first compare the proposed (temperature-based, margin-based and angle-based) logit calibrations with the original CE, Po+Trip [15], and Logit [30] in the single-model transfer task. In this task, we take one surrogate model for training, and test the targeted transferability in attacking other 3 models.

As shown in Table 1, the original CE loss function produces a worst performance than the Po+Trip and Logit. But after performing the logit calibration in the CE loss function, we can find a significant performance boost compared with the original CE. All the calibration methods can outperform the Logit, especially when using the ResNet50 and Dense121 as the surrogate. These results indicate that the logit margin can significantly influence the performance of the targeted transferability. On the other aspect, we find that $T = 10$ has better performance than $T = 5$ on the VGG-16, suggesting that different models may need different $T$. Instead of finding the best $T$ for a different model, the Margin-based calibration can solve the issue and reach the overall best transferability in all four models. However, we find that the Angle-based calibration is not working on the VGG16, which needs further investigation.

## 4.2 The Influence of Different $T$ in CE

In this section, we evaluate the influence of using different T in the CE loss function. The results are reported in Table 2. From the Table, we can have the following observations. **1)** The scaling factor $T$ has a significant influence on the targeted transferability. Specifically, there is a large decrease in performance when using a small $T = 0.5$. After increasing the $T$, we can observe the number of successfully attacked samples will increase. **2)** The optimal $T$ for different model is different. For example, $T = 5$ can produce the overall best performance for ResNet50, Dense121, and Inception v3, while the VGG16 with fewer convolutional layers requires a large $T$ to obtain better transferability. **3)** The performance are comparable when using $T = 5$ and $T = 10$ for ResNet50, Dense121, and Inception v3. This is because that we use I-FSGM for optimization, which only considers the sign of the gradients. **4)** Using a larger $T$, the performance will be similar to the Logit loss function (see Table 1). We provide a deep analysis of this phenomenon in the supplementary material.

[2]https://github.com/ZhengyuZhao/Targeted-Tansfer

Table 2: The targeted transfer success rates (%) by using different $T$ in CE loss function. (Results with 20/100/300 iterations are reported.)

| Attack | Surrogate Model: ResNet50 | | | Surrogate Model: Dense121 | | |
|---|---|---|---|---|---|---|
| | →Dense121 | →VGG16 | →Inc-v3 | →Res50 | →VGG16 | →Inc-v3 |
| T=0.5 | 13.2/16.0/19.5 | 7.1/9.5/11.0 | 1.2/1.8/2.4 | 4.2/5.0/6.2 | 2.5/3.5/3.2 | 0.6/0.9/1.1 |
| T=1 | 27.0/40.2/42.7 | 17.4/27.6/29.1 | 2.3/4.1/4.6 | 12.3/17.2/18.4 | 8.6/10.5/10.9 | 1.6/2.3/2.8 |
| T=2 | 34.2/62.8/67.7 | 24.4/52.3/53.9 | 3.3/7.2/8.5 | 18.7/35.0/36.1 | 13.2/27.3/27.0 | 2.2/5.5/6.1 |
| T=5 | 33.3/69.9/**77.8** | 24.8/59.9/66.1 | 3.1/9.4/**12.2** | 19.3/43.4/47.5 | 14.6/36.6/39.4 | 2.3/7.3/8.8 |
| T= 10 | 31.6/68.5/77.0 | 23.6/58.5/**66.4** | 2.8/9.4/11.6 | 17.9/43.2/**49.3** | 13.4/36.8/**41.5** | 2.2/7.7/**8.8** |
| T = 20 | 30.4/65.6/74.3 | 22.9/55.4/63.6 | 3.2/9.0/11.6 | 17.6/40.3/46.2 | 13.4/35.4/40.1 | 2.3/6.7/8.7 |

| Attack | Surrogate Model: VGG16 | | | Surrogate Model: Inc-v3 | | |
|---|---|---|---|---|---|---|
| | →Res50 | →Dense121 | →Inc-v3 | →Res50 | →Dense121 | →VGG16 |
| T=0.5 | 0.2/0.1/0.2 | 0.1/0.1/0.1 | 0/0/0 | 0.3/0.9/0.9 | 0.3/0.8/1.4 | 0.3/0.6/1.3 |
| T=1 | 0.5/0.3/0.6 | 0.6/0.3/0.3 | 0/0/0.1 | 0.7/1.2/1.8 | 0.6/1.3/1.9 | 0.4/0.8/1.3 |
| T=2 | 1.6/1.8/1.8 | 1.8/1.9/1.6 | 0.2/0.2/0.2 | 0.6/1.5/2.0 | 0.4/1.7/2.2 | 0.5/1.2/2.0 |
| T=5 | 3.1/7.0/6.9 | 3.3/7.6/7.8 | 0.2/0.9/0.8 | 0.7/1.7/2.1 | 0.5/1.9/3.3 | 0.4/1.6/2.6 |
| T= 10 | 3.6/9.0/9.7 | 3.4/10.5/11.7 | 0.3/1.1/1.3 | 0.5/1.3/1.9 | 0.6/2.0/2.7 | 0.4/1.5/**2.8** |
| T = 20 | 3.4/9.7/**11.1** | 3.6/12.7/**13.8** | 0.3/1.2/**1.3** | 0.5/1.4/**2.3** | 0.6/1.8/**3.1** | 0.5/1.6/2.4 |

## 4.3 The Targeted Success rates for Transfer with Varied Targets

In Table 3, we report the result of a worse-case transfer scenario by gradually varying the target class from the highest-ranked to the lowest one, and have the following findings: **1)** The three types of logit calibration methods can improve the targeted transfer success rate over the original CE. The angle-based calibration has the best performance. But, we notice that the margin-based calibration doesn't work well in this setting. **2)** The Temperature-based (T=5/10) and the Angle-based calibrations can outperform the Logit loss by a large margin, especially the Angle-based calibration.

## 4.4 The Mutual Benefits of Different Calibration Methods

In this part, we evaluate the mutual benefits of combining different calibrations and can have the following findings. **1)** Combining the T=5/10/20 and Margin, there is no increase in performance compared with using one of them. This is because that the gradient directions of these two methods are very similar. **2)** Combining the T=5 and Angle, we can observe a further improvement when using ResNet50 and Dense121 as the surrogate model, *e.g.*, the transferable rate of "ResNet50 → Dense121" is increased to 82.4% with 300 iterations. Since the Angle obtains poor performance on

Table 3: Targeted transfer success rate (%) when varying the target from the high-ranked class to low.

| | 2nd | 10th | 200th | 500th | 800th | 1000th |
|---|---|---|---|---|---|---|
| Logit | 83.7 | 83.2 | 74.5 | 71.5 | 64.9 | 52.4 |
| CE | 77.4 | 58.6 | 26.9 | 23.7 | 16.7 | 7.0 |
| CE/5 | 91.3 | 88.7 | 77.1 | 75.8 | 70.1 | 58.8 |
| CE/10 | 89.0 | 87.8 | 81.0 | 79.2 | 73.5 | 62.5 |
| Margin | 87.4 | 81.7 | 61.3 | 51.6 | 43.1 | 23.0 |
| Angle | 92.4 | 89.1 | 80.3 | 79.2 | 76.1 | 66.3 |

VGG16, the transferable rates of corresponding combinations are also low in T=5/10+Angle, but T=20+Angle can deal with this issue. **3)** Combining the Margin and Angle, there are only slight improvements on ResNet50 and Dense121, while it can alleviate the negative effects caused by the angle-based calibration. Finally, by jointly considering the results in Table 1, 2 and 4, we suggest using T=5 + Angle for CNNs with more layers and the single Margin-based calibration for CNNs with fewer layers to achieve better targeted transfer attack.

## 4.5 Comparison with The TTP Method

In this section, we further evaluate the proposed temperature-based logit calibration in the GAN-based targeted attacks. Following the setting in TTP [21], we sampled 50K images from the ImageNet training set and 50K images from the Painting dataset[3], which are used to train the targeted generators from different source domains. Instead of using the distribution matching and neighborhood similarity matching loss [21], we only use the cross-entropy function for training the targeted generators while keeping other settings identical. More training and evaluation details used by TTP can be referred to [21]. We used the ResNet50 as the surrogate model and reported the results in Table 5.

---

[3] https://www.kaggle.com/c/painter-by-numbers

Table 4: The comparison of combining logit calibrations. (The targeted transfer success rates (%) with 20/100/300 iterations are reported.)

| Attack | Surrogate Model: ResNet50 | | | Surrogate Model: Dense121 | | |
|---|---|---|---|---|---|---|
| | →Dense121 | →VGG16 | →Inc-v3 | →Res50 | →VGG16 | →Inc-v3 |
| T=5 + Margin | 33.8/69.8/77.2 | 24.0/59.0/65.5 | 3.3/9.6/11.1 | 19.3/44.3/47.8 | 14.1/37.7/40.8 | 2.5/7.5/9.4 |
| T=5 + Angle | 34.5/74.3/**82.4** | 25.6/66.5/**72.2** | 3.6/10.5/**13.1** | 20.3/52.7/**61.9** | 15.8/45.0/**53.6** | 2.3/9.2/**12.7** |
| T=10 + Margin | 32.7/69.5/77.3 | 22.8/59.4/66.3 | 12.9/9.7/11.5 | 18.3/44.1/49.1 | 13.7/36.9/41.6 | 2.4/8.3/9.2 |
| T=10 + Angle | 33.0/69.8/79.1 | 24.4/59.0/68.9 | 3.4/10.0/12.9 | 19.4/47.2/56.1 | 14.8/40.1/47.0 | 2.5/8.3/11.0 |
| T=20 + Margin | 33.0/69.2/76.2 | 23.1/58.4/65.8 | 3.2/9.5/11.8 | 19.1/43.4/48.5 | 13.9/36.7/41.4 | 2.4/7.8/9.5 |
| T=20 + Angle | 34.2/68.6/76.5 | 24.7/58.7/66.6 | 3.4/9.7/12.7 | 20.0/44.4/50.9 | 15.5/38.4/43.7 | 2.5/8.2/9.5 |
| Margin + Angle | 34.4/70.8/78.1 | 24.3/60.2/67.4 | 3.5/10.4/12.6 | 19.9/46.6/52.7 | 15.2/39.3/44.5 | 2.7/8.2/9.9 |

| Attack | Surrogate Model: VGG16 | | | Surrogate Model: Inc-v3 | | |
|---|---|---|---|---|---|---|
| | →Res50 | →Dense121 | →Inc-v3 | →Res50 | →Dense121 | →VGG16 |
| T=5 + Margin | 3.5/10.2/11.4 | 3.7/12.4/14.6 | 0.3/1.1/1.3 | 0.5/1.4/1.6 | 0.6/2.1/2.9 | 0.5/1.7/2.8 |
| T=5 + Angle | 2.2/2.5/2.3 | 2.4/2.6/2.3 | 0.2/0.1/0.2 | 0.5/1.6/**2.4** | 0.6/2.0/3.1 | 0.5/1.7/2.5 |
| T=10 + Margin | 3.2/10.7/11.7 | 3.4/12.9/**15.0** | 0.2/1.0/**1.4** | 0.5/1.4/1.9 | 0.5/1.9/3.0 | 0.3/1.5/2.3 |
| T=10 + Angle | 3.4/6.2/5.1 | 3.5/7.5/7.0 | 0.2/0.6/0.6 | 0.6/1.3/1.9 | 0.6/2.0/3.2 | 0.5/1.6/2.6 |
| T=20 + Margin | 3.5/10.1/**11.8** | 3.4/12.0/14.9 | 0.3/1.2/**1.4** | 0.6/1.2/1.9 | 0.5/1.9/2.9 | 0.5/1.6/2.7 |
| T=20 + Angle | 3.2/9.7/10.1 | 3.9/11.9/13.3 | 0.3/1.0/1.2 | 0.6/1.6/2.0 | 0.6/2.0/**3.5** | 0.5/1.7/**2.9** |
| Margin + Angle | 3.3/9.8/11.1 | 3.5/12.6/14.6 | 0.3/1.2/**1.4** | 0.6/1.4/2.0 | 0.6/1.7/3.1 | 0.5/1.5/2.6 |

Table 5: **Comparison with TTP [21] on Target Transferablity.** The averaged Top-1 targeted accuracy (%) across 10 targets are computed with 49.95K ImageNet validation samples. Perturbation budget: $l_\infty \leq 16$. * indicates the training surrogate model.

| Dataset | Loss | ResNet50* | VGG19$_{BN}$ | Dense121 | ResNet152 | WRN-50-2 | Average |
|---|---|---|---|---|---|---|---|
| ImageNet | TTP | 97.02* | 78.15 | 81.64 | 80.56 | 78.25 | 83.12 |
| | CE | 97.15* | 70.44 | 78.96 | 76.22 | 78.24 | 80.20 |
| | CE (T=5) | **99.18*** | **86.65** | **90.55** | **90.30** | **93.22** | **91.98** |
| Painting | TTP | 96.63* | 73.09 | 84.76 | 76.27 | 75.92 | 81.33 |
| | CE (T=5) | **98.95*** | **82.97** | **87.07** | **87.81** | **91.70** | **89.70** |

From Table 5, we make the following findings. **1)** By using ImageNet as the training dataset, the TTP shows better transferability than the CE in attacking other black-box models. The average targeted accuracy of TTP is around 3% higher than that of CE. **2)** After downscale the logit by 5 in the CE loss function (CE (T=5)), we can observe a significant boost of the Top-1 targeted accuracy for all models, reaching the average targeted accuracy of 91.98% (ImageNet). **3)** For both ImageNet and Painting as the training source, the CE (T=5) can surpass the TTP by a large margin (91.98% vs. 83.12% & 89.70% vs. 81.33%). These experimental results demonstrate that the proposed temperate-based logit calibration is also effective in training generator-based targeted attackers. Note that, compared to TPP, our logit calibration has the benefit of without using any data from the target class.

## 5 Conclusion

In this study, we analyzed the logit margin in different loss functions for the transferable targeted attack, and find that the margin will quickly get saturated in the CE loss and thus limited the transferablity. To deal with this issue, we introduce to use logit calibrations in the CE loss function, including Temperature-based, Margin-based, and Angle-based. Experimental results verified the effectiveness of using the logit calibration in the CE loss function for crafting transferable targeted adversarial samples. The proposed logit calibration methods are simple and easy to implement, which can achieve state-of-the-art performance in transferable targeted attack.

**Potential Social Impact**. Our findings in targeted transfer attacks can potentially motivate the AI community to design more robust defenses against transferable attacks. In the long run, it may also be directly used for suitable social applications, such as protecting privacy. Contrariwise, some applications may use targeted transferable attacks in a harmful manner to damage the outcome of AI systems, especially in scenarios of speech recognition and facial verification systems. Finally, we firmly believe that our investigation in this study can provide valuable insight for future researchers by using the logit calibration for both adversarial attack and defense.

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
