# Supplementary material

## 1  Logit vs. Cross-Entropy with large $T$

In this part, we analysis the relation between the Logit and the Cross-Entropy Loss function. The Logit loss function is:

$$L_{Logit} = -z_t, \tag{1}$$

where $-z_t$ denotes the logit value of the target class $t$. Then we can have the gradient wrt. input feature $\phi(\hat{x})$ as:

$$\frac{\partial L_{Logit}}{\partial \phi(\hat{x})} = -W_t. \tag{2}$$

The Cross-Entropy loss function with $T$ is:

$$L_{CE}^T = -\log(\hat{p}_t), \tag{3}$$

where $\hat{p}_t = \frac{e^{z_t/T}}{\sum e^{z_i/T}}$. We can compute the gradient wrt. input $\phi(\hat{x})$ as:

$$\frac{\partial L_{CE}^T}{\partial \phi(\hat{x})} = \sum_i -\hat{p}_t \frac{(W_t - W_i)}{T}. \tag{4}$$

When using a large $T$, the distribution $\hat{\boldsymbol{p_i}}$ will be extremely smooth over different classes. And we can get the $\hat{p}_i \approx \frac{1}{N}$ for each class, where $N$ is the number of classes. In this study, we conduct experiments on the ImageNet dataset ($N = 1000$), then Eq. 4 will become:

$$\begin{aligned} \frac{\partial L_{CE}^T}{\partial \phi(\hat{x})} &\approx \sum_i -\frac{(W_t - W_i)}{NT} \\ &\approx -\frac{W_t}{T} + \frac{1}{NT}\sum_i W_i \\ &\approx -\frac{W_t}{T}, \end{aligned} \tag{5}$$

which is approximate $\frac{1}{T}$ of the gradient in Eq 2. On the other aspect, the I-FGSM is used for optimization,

$$\hat{x}_{i+1} = \hat{x}'_i + \alpha \cdot \text{sign}(\nabla_{\hat{x}} J(\hat{x}'_i, y)) \tag{6}$$

which only considers the Sign of the gradient. Therefore, the Eq 2 and Eq 5 will update the perturbation in a similar direction.

Based on the above analysis, we can consider the Logit loss function as a special case of Cross-Entropy when using a large $T$. In Figure 1 and Table 1, we compare the performance of the Logit and CE (T=50 & T=100). From the Figure and the Table, we can find that the performance of the Logit and CE (T=50 & T=100) is very similar. These results verify our analysis of the relation between the Logit and the CE with large $T$.

36th Conference on Neural Information Processing Systems (NeurIPS 2022).

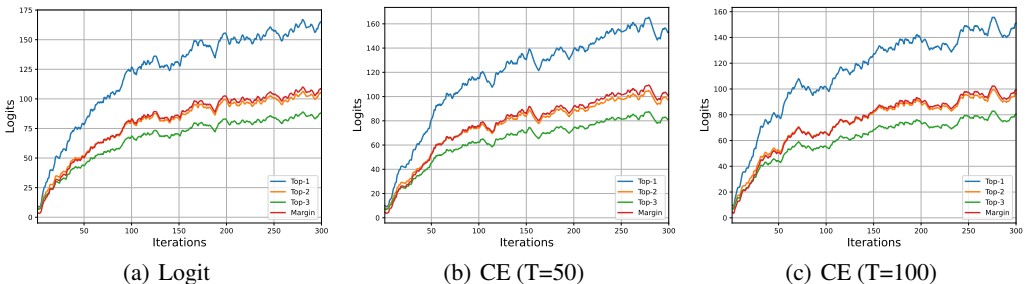

|          | (a) Logit | (b) CE (T=50) | (c) CE (T=100) |
|----------|-----------|---------------|----------------|

Figure 1: The average Top-3 logits and logit margin of 50 adversarial samples trained by the Logit, CE (T=50) and CE (T=100) loss functions for crafting the ResNet-50.

Table 1: Comparing the Logit with CE (T=50 & T=100) in the single-model transfer scenario. (The targeted transfer success rates (%) with 20/100/300 iterations are reported).

| Attack | Surrogate Model: ResNet50 | | | Surrogate Model: Dense121 | | |
|--------|→Dense121|→VGG16|→Inc-v3|→Res50|→VGG16|→Inc-v3|
| Logit  | 31.4/64.0/71.8 | 23.8/55.0/62.4 | 3.1/8.6/10.9 | 17.4/38.6/43.5 | 13.7/33.8/37.8 | 2.3/6.6/7.5 |
| T=50   | 30.2/64.7/72.7 | 23.3/55.1/62.9 | 2.9/8.8/11.4 | 17.3/39.6/44.8 | 12.7/34.3/38.3 | 2.4/6.7/8.3 |
| T= 100 | 30.0/64.7/72.3 | 22.8/54.4/61.9 | 3.1/8.7/10.7 | 17.0/39.7/44.7 | 13.0/33.7/39.1 | 2.2/6.5/8.1 |

| Attack | Surrogate Model: VGG16 | | | Surrogate Model: Inc-v3 | | |
|--------|→Res50|→Dense121|→Inc-v3|→Res50|→Dense121|→VGG16|
| Logit  | 3.4/9.9/11.6 | 3.5/12.0/13.9 | 0.3/1.0/1.3 | 0.6/1.1/2.0 | 0.6/1.9/3.0 | 0.6/1.5/2.8 |
| T=50   | 3.1/10.2/11.4 | 3.9/12.0/14.5 | 0.1/1.1/1.3 | 0.6/1.8/2.1 | 0.6/2.0/3.0 | 0.3/1.7/2.7 |
| T= 100 | 3.6/9.8/11.3 | 3.4/11.8/13.9 | 0.4/1.2/1.4 | 0.6/1.6/2.0 | 0.4/2.1/3.0 | 0.4/1.7/2.8 |

## 2   The probabilities in Margin-based calibration

In the Margin-based calibration, we calibrate the logits by using the margin between the Top-2 logits in each iteration. The calibrated logits is:

$$\tilde{z}_i = \frac{z_i}{\hat{z}_1 - \hat{z}_2}. \tag{7}$$

where $\hat{z}$ represents the sorted logits. Suppose the $\tilde{z}$ is sorted, the Top-1 logit $\tilde{z}_1$ will be the target class $\tilde{z}_t$ after a few iterations. Therefore, the corresponding calibrated probability of the target class will be:

$$
\begin{aligned}
p_t &= \frac{1}{1 + \sum_{i \neq t} e^{-(\tilde{z}_t - \tilde{z}_i)}} \\
&= \frac{1}{1 + e^{-(\tilde{z}_t - \tilde{z}_i)} + \sum_{i=2} e^{-(\tilde{z}_t - \tilde{z}_i)}} \\
&= \frac{1}{1 + e^{-\frac{\hat{z}_t - \hat{z}_2}{\tilde{z}_t - \tilde{z}_2}} + \sum_{i=2} e^{-(\tilde{z}_t - \tilde{z}_i)}} \\
&< \frac{1}{1 + e^{-1}}.
\end{aligned}
\tag{8}
$$

Correspondingly, we can have the probability of $1 - p_t > 1 - \frac{1}{1+e^{-1}}$. Therefore, the probability $p_{\hat{1}}$ of the Top-1 non-target class will be larger than the average probability of all non-target classes, denoted as:

$$p_{\hat{1}} = \frac{1}{e^{\tilde{z}_{\hat{1}} - \tilde{z}_t} + \sum_{i \neq t} e^{\tilde{z}_i - \tilde{z}_{\hat{1}}}} > \frac{1}{N-1}(1 - \frac{1}{1+e^{-1}}). \tag{9}$$

Therefore, our Margin-based calibration can adaptively deal with the vanishing gradient issue in the original CE loss function.

## 3 More Experimental Results

### 3.1 The targeted transfer success on using ResNet-18 as the surrogate model

In Table 1 of the main manuscript, we can find that a large "T" is preferred to achieve better performance in the VGG16 when using the Margin-based calibration. We guess the main reason for this phenomenon is mainly due to the influence of model depth. For the CNN models with fewer layers, a large normalization factor "T" is preferred to achieve higher targeted transferability. In our Margin-based calibration, the denominator "T" (logit margin between the first and second logits) will keep increasing along with the optimization iterations and thus leads to better performance.

To further check the influence of model depth, we leverage the ResNet-18 with fewer convolution layers as the surrogate model and reported the results in the following Table 2. We also find that a large T can achieve better performance in the margin-based calibration. These results may suggests that a large "T" is preferred to CNNs with few layers.

Table 2: The targeted transfer success rate (%) with the ResNet-18 as the surrogate model.

| Attack | Inc-v3 | ResNet-50 | Dense-121 | VGG-16 |
|---|---|---|---|---|
| CE | 2.1/3.0/3.0 | 19.2/24.0/26.0 | 18.6/24.0/24.6 | 15.9/19.3/19.0 |
| CE/5 | 3.9/10.8/11.9 | 27.8/60.7/63.6 | 27.2/57.5/61.6 | 23.7/53.0/56.6 |
| CE/10 | 3.6/11.2/13.2 | 25.9/59.7/66.9 | 25.9/57.2/64.2 | 22.2/53.0/59.7 |
| CE/20 | 3.9/11.4/13.0 | 25.2/57.8/64.2 | 24.8/54.3/60.7 | 21.1/49.7/57.1 |
| Margin | 4.1/11.3/13.1 | 27.3/60.1/65.3 | 27.3/57.3/62.9 | 23.4/53.5/58.6 |
| Angle | 3.7/8.2/8.4 | 27.1/51.5/54.3 | 28.1/52.8/55.7 | 23.9/44.9/46.2 |
| Logits | 3.7/10.0/12.2 | 24.8/55.6/60.7 | 24.3/53.6/58.5. | 21.2/49.4/54.9 |

### 3.2 Transfer-based attacks on Google Cloud Vision

Following the evaluation protocol in [2], we randomly select 100 images to conduct a real-world adversarial attack on the Google Cloud Vision API. The attacking performance is computed based on transfer-based attacks of the ensemble of four CNNs (i.e., Inc-V3, ResNet-50, Dense-121 and VGG-16). The results are shown in Table 3. We can find that the results of the Logit and CE (T=5) are very similar. But the Margin-based calibration performs worse than Logit and CE (T=5). These results reveal that our logit calibration-based targeted transfer attacks can potential cause a thread to the real-world Google Cloud Vision API.

Table 3: Non-targeted and targeted transfer success rates (%) on Google Cloud Vision API.

| | Logit | CE (T=5) | Margin |
|---|---|---|---|
| Targeted | 16 | 15 | 12 |
| Non-targeted | 51 | 53 | 42 |

### 3.3 The effective of using logit calibration in non-targeted attack

We further conducted the experiments on the CIFAR-10 dataset under the untargeted attack setting based on the code provided by [1]. The ResNet-18 is used as the white-box model for crafting the perturbation by training with the I-FGSM for 20 iterations. The DenseNet, GoogLeNet and SENet18 are black-box models. Table 1 reported the fooling rate of attacking the 10,000 images in the CIFAR-10 testing set.

From Table 4, we can find that the fooling rate continually increases along with the T in the white-box attack. In transfer black-box attacks, the best fooling rates are obtained at T=5 or T=10, and the fooling rate will decrease when further increases T. These results also can validate the effectiveness of logit calibration in non-targeted attacks on a small dataset.

## References

[1] Qian Huang, Isay Katsman, Horace He, Zeqi Gu, Serge Belongie, and Ser-Nam Lim. Enhancing adversarial example transferability with an intermediate level attack. In *ICCV*, 2019.

Table 4: The transfer untargeted fooling rate of training with ResNet-18 and testing by the DenseNet-121, GoogleNet and SENet-18 on CIFAR-10.

|         | ResNet-18* | DenseNet-121 | GoogLeNet | SENet-18 |
|---------|------------|--------------|-----------|----------|
| T=0.5   | 89.77      | 50.23        | 37.43     | 51.04    |
| T=1     | 91.61      | 50.78        | 37.30     | 51.20    |
| T=2     | 91.39      | 51.14        | 37.60     | 51.65    |
| T=5     | 92.01      | 55.56        | 41.77     | 55.74    |
| T=10    | 94.04      | 54.76        | 42.41     | 55.10    |
| T=20    | 94.20      | 53.33        | 41.31     | 54.11    |

[2] Zhengyu Zhao, Zhuoran Liu, and Martha Larson. On success and simplicity: A second look at transferable targeted attacks. *NeurIPS*, 34, 2021.