# OpenReview forum: "Logit Margin Matters: Improving Transferable Targeted Adversarial Attack by Logit Calibration"
_NeurIPS.cc/2022/Conference — NeurIPS 2022 Submitted_

### Official Review · Reviewer_WQZv · 2022-07-07

**Rating:** 5
**Confidence:** 4
**Soundness:** 2 fair
**Presentation:** 3 good
**Contribution:** 2 fair

**Summary:**

The paper targets improving the transferability of adversarial attack using the logit calibration. Despite the recent success in untargeted black-box attacks, the targeted transferability of adversarial attacks remains challenging. The paper takes a closer look at the vanishing gradient issue in the CE loss function which is commonly used to learn transferable adversarial samples and suggests that the logit margin between the targeted and non-targeted classes quickly gets saturated during the optimization process. So, to improve transferability they aim to enlarging logit margins which consequently reduce saturation and  enable longer optimization and iterations. The paper investigates three different types of logit calibrations including temperature-based, angle-based and margin-based inspired by previous studies and techniques.  Experiment conducted using ImageNet dataset and different methods including ResNet50, DenseNet-121, VGG-16 and Inception-v3. Results are compared to SOTA methods including Po+Trip, Logit, and TTP.

**Questions:**

- There is a large novelty overlap between [30] and the current method. It is proper to get the similarity and distinction discussed upfront. Specifically,  discussion around CE starting line 36 is getting very confusing and blurry going through line 56. Vanishing gradient and the use of Logit loss has been discussed and proposed in previous arts, for example [30]. This gets discussed later at line 145+ and in method, however, I feel the contribution of the current work can get discussed in a more clear way and upfront in introduction.

Experiments are limited and can get improved:
- First the organization of the results is not optimized or ideal. (1) Following Table 1, 2 and 3 is very hard and replacing this with line graphs that capture progress through iterations would be very beneficial. (2) Overall collective of Table 1, 2, 3 seems to be more exploratory and ablation tables rather than the main results. The main message that I read from these two tables are T=10, 20 and a combination of Margin + Angle or T + Angle can result in the best outcome. So, why not introduce a single best receipt and present everything else as ablations? (3) Also I would suggest sticking with conventional methods such as heat-map to summarized heavy tables, as it is a norm in adversarial attack literature.  [21] have good examples of result presentations.

- One relevant question, is also as Table 2 suggests the best outcome is coming from T=10 or T=20 so why in Table 3 we analyze the effect of combining logits as T=5 +Margin or Angle? If any underlying study suggests this, results should be provided.

The results could be strengthened by:
- Providing experiments on another dataset (e.g., CIFAR-10, MNIST, SVHN) Since the proposed method works well on ImageNet, it could only be a minor concern.
- Incorporating study of a real-world attack for example the Google Cloud Vision API.
- Providing targeted success rates for transfer with varied targets.


**Limitations:**

I cannot find any specific discussion around the potential negative social impact of this work. Also, the limitation of the method was not addressed in the paper.

To improve this part, discussion around the benefits of adversarial attack research can get discussed. Potentially this can motivate the AI community to design stronger defenses against transferable attacks, and  in the long run such results can be directly used for social good applications, such as protecting privacy. On the contrary, there are applications that can benefit from transferable attacks in a harmful manner to damage the outcome of any AI system, e.g. imaging a scenario that someone uses such attacks to intrude with the outcome of a medical AI device.

I also suggest that authors discuss the limitation of the work, failure cases and processing time.

Overall, I enjoy reviewing this paper and looking forward to reading the authors' responses.


**Strengths And Weaknesses:**

Strength:
- Quality: The writing quality is very good, very easy to follow, there are areas that can be improved but nothing major.
- Clarity: Very clear. Easy to understand the motivation and the thought process of different method’s component.
- Significance and novelty: Novelty is a bit limited and built up mostly on top of previous methods,  but it is also not a weakness because this work attempts to solve an interesting problem and the analysis and results are valuable. The results are somewhat important. Mostly inspired by Logit [30], future researchers might use the suggested logit calibration and increase the number of iterations for targeted attacks.

Weaknesses:
- The quality of results and presentation of it could be improved significantly. (see below)
- The distinction between [30] and this paper should be clarified and the introduction in page 2 [line 36-56] can benefit from a re-writing. (see below)

---

> ### Author Response · Authors · 2022-08-02
> **Response to Reviewer WQZv**
>
> Thanks for your comments and valuable suggestions for the presentation. We would like to address your concerns in the following aspects.
>
> >**Comment 1:** Clarified the distinction between [30] in the introduction.
> >
> >**Response 1:** Thanks for your suggestion. We will rewrite this part in the introduction to better clarify the contributions of this study.
>
> >**Comment 2:** To have a better presentation of the tables.
> >
> >**Response 2:** We agree that a better presentation of the tables is needed. Currently, we use tables instead of line graphs that capture progress through iterations mainly due to the results of different calibration methods being very similar, and their lines will largely overlap with others. In the revision, we will report the average targeted transfer success rate with 3digits at most for Tables 1 & 3 instead of the average number of successfully attacked samples with 4 digits, and replace Table 2 with line graphs.
>
> >**Comment 3:** Why not introduce a single best receipt and present everything else as ablations?
> >
> >**Response 3:** The main reasons are: **(1)** The primary goal is solving the saturated issue in the CE loss for learning better transferable targeted adversarial attacks. Therefore, we first evaluate the effectiveness of different logit calibrations. **(2)** Since the logit calibration works, we then test their mutual effects by combining them jointly. However, we find that the optimal combination is different for different models, and there isn't a universal receipt for them. Consequently, we didn't introduce a single best receipt and present else as ablations.
> >
> >On the other aspect, based on the results in the manuscript and new results in the rebuttal, we might suggest using (T=5 + Margin) or (T=5 + Angle) for CNNs with more layers and the single Margin-based calibration for CNNs with fewer layers.
>
> >**Comment 4:** Why $T=5$ +Margin or Angle in Table 3?
> >
> >**Response 4:** We currently use the same $T=5$ mainly based on the result of ResNet-50, instead of the optimal $T$ for each model. During the rebuttal, we test the performance of $T=10,20$ + (Margin or Angle). The results are reported in Table 1. We can find that the $T$ has a marginal influence in the combination of "$T$ + Margin", while largely increasing the performance of "$T$ + Angle" of VGG16.
>
> **Table 1.** The comparison of combining logit calibration.
>
> (1) Surrogate model: **ResNet-50**
> |       | Dense121  | VGG16     | Inc-v3     |
> | -     | :-:       | :-:        | :-:        |
> |T=5 + Margin  |338.2/698.4/772    |239.6/590/655.4   |33.4/96/111      |
> |T=5 + Angle   |345.2/742.6/823.8  |256.2/664.8/721.6 |35.8/104.6/131.4 |
> |T=10 + Margin |326.8/694.6/772.8  |227.8/593.6/663.2 |129.4/96.8/114.6 |
> |T=10 + Angle  |329.6/697.6/790.6  |244.2/590.2/689.4 |33.6/99.8/128.6  |
> |T=20 + Margin |330/691.6/762.4    |230.8/584.4/658.2 |31.6/95/117.8    |
> |T=20 + Angle  |342.2/686.2/764.6  |247.4/587/666.2   |34.4/97.4/126.8  |
> |Margin+Angle  |344/708.4/781.4    |242.6/601.8/673.8 |35/103.6/125.8   |
>
> (2) Surrogate model: **Denss121**
> |       | ResNet50  | VGG-16     | Inc-v3     |
> | -     | :-:       | :-:        | :-:        |
> |T=5 + Margin |192.6/442.6/477.8  |141.2/377.4/408.4 |25.4/74.8/93.6 |
> |T=5 + Angle  |202.6/526.6/619.2  |158.2/450.2/536.4 |23.4/92/127.2  |
> |T=10 + Margin|183.2/441.4/491.2  |136.6/369.4/416.4 |24.4/82.8/91.8 |
> |T=10 + Angle |193.8/472/561.2    |148.2/400.6/470.2 |25.2/82.8/109.8|
> |T=20 + Margin|191/433.8/485.4    |138.8/366.6/414.2 |23.6/78.2/95.4 |
> |T=20 + Angle |199.6/443.8/508.6  |155.4/383.8/437   |24.6/82.4/95.2 |
> |Margin+Angle |198.8/465.6/527.4  |152.2/392.8/445.4 |27/82/99       |
>
> (3) Surrogate model: **VGG16**
> |       | ResNet50    | Dense121   | Inc-v3     |
> | -     | :-:       | :-:         | :-:        |
> |T=5 + Margin  |34.8/101.8/114    |37.2/123.6/145.6 | 3/10.8/13    |
> |T=5 + Angle   |21.6/25/23.4      |23.6/25.6/23.2   | 1.6/1.4/1.6  |
> |T=10 + Margin |31.6/107.2/117.4  |34.4/129.4/149.6 | 2.2/10.4/14.2|
> |T=10 + Angle  |34/62/50.8        |34.8/75/70.2     | 2.4/6.4/5.8  |
> |T=20 + Margin |34.6/100.8/118.2  |33.6/120.2/148.6 | 2.8/12.4/14.4|
> |T=20 + Angle  |32.4/96.6/101     |38.8/119.4/133.2 | 2.8/10.4/12  |
> |Margin+Angle  |33/98.4/111.2     |35.4/126.4/146   | 2.6/12/14    |
>
> (4) Surrogate model: **Inc-v3**
> |       | Dense121  | VGG-16     | Inc-v3     |
> | -     | :-:       | :-:        | :-:        |
> |T=5 + Margin |4.8/14.4/16    |6.2/21.2/28.6 | 5/17.2/28.4  |
> |T=5 + Angle  |5.2/16/20.4    |5.8/19.6/31.2 | 5.4/16.6/24.6|
> |T=10 + Margin|5.4/14/19.2    |4.6/18.8/30.4 | 3.2/14.8/23  |
> |T=10 + Angle |5.8/13.4/18.6  |6/19.6/32.2   | 4.6/16.4/25.6|
> |T=20 + Margin|6.4/12.2/19.4  |5/19/29       | 4.8/16.4/26.8|
> |T=20 + Angle |6.4/16.2/20.4  |5.6/19.6/35.2 | 4.8/17/28.8  |
> |Margin+Angle |6.4/14/21      |5.6/17/31.2   | 5.4/15.4/26.2|
>
> >**Comment 5:** Limitation of this study.
> >
> >**Response 5:** Thanks for your valuable suggestions. We will add this information in the revision.

---

> > ### Author Response · Authors · 2022-08-02
> > **Results on another dataset, attacking Google API, and transfer with varied targets**
> >
> > >**Exp 1:** Experiments on another dataset (e.g., CIFAR-10, MNIST, SVHN)
> > >
> > >**Response 1:**  During the rebuttal, we conducted the experiments on the CIFAR-10 dataset under the untargeted attack setting based on the code provided by [a]. The ResNet-18 is used as the white-box model for crafting the perturbation by training with the I-FGSM for 20 iterations. The DenseNet, GoogLeNet and SENet18 are black-box models. Table 1 reported the fooling rate of attacking the 10,000 images in the CIFAR-10 testing set.
> > >
> > >From Table 1, we can find that the fooling rate continually increases along with the T in the white-box attack. In transfer black-box attacks, the best fooling rates are obtained at T=5 or T=10, and the fooling rate will decrease when further increases T. These results also can validate the effectiveness of logit calibration in non-targeted attacks on a small dataset.
> > >
> > >[a] Enhancing Adversarial Example Transferability with an Intermediate Level Attack, *ICCV 2019*.
> >
> > Table 1: The transfer untargeted fooling rate of training with ResNet-18 and testing by the DenseNet-121, GoogleNet and SENet-18 on CIFAR-10.
> > |       | ResNet-18*| DenseNet-121 | GoogLeNet   | SENet-18 |
> > | -     | :-:  | :-:  | :-:  |:-:   |
> > |T=0.5  |89.77 |50.23 |37.43 |51.04 |
> > |T=1    |91.61 |50.78 |37.30 |51.20 |
> > |T=2    |91.39 |51.14 |37.60 |51.65 |
> > |T=5    |92.01 |55.56 |41.77 |55.74 |
> > |T=10   |94.04 |54.76 |42.41 |55.10 |
> > |T=20   |94.20 |53.33 |41.31 |54.11 |
> >
> > >**Exp 2:** A real-world attack on the Google Cloud Vision API.
> > >
> > >**Response 6:** We randomly select 100 images and compute the attacking performance of the ensemble of four CNNs using the same evaluation protocol in [30]. The results are as follows. We can find that the results of the Logit and CE (T=5) are very similar. But the Margin-based calibration performs worse than Logit and CE (T=5).
> >
> > Table 2: Non-targeted and targeted transfer success rates (%) on Google Cloud Vision.
> > |       | Logit | CE (T=5)   | Margin |
> > | -     | :-:  | :-:  |  :-:   |
> > | Targeted    |  16| 15  | 12 |
> > | Non-targeted| 51 | 53 | 42 |
> >
> > >**Exp 3:** The targeted success rates for transfer with varied targets.
> > >
> > >**Response 3:** The targeted transfer success rate with varied targets is reported in Table 3, and we can have the following findings. **(1)** The three types of logit calibration methods can improve the targeted transfer success rate over the original CE. The angle-based calibration has the best performance. But, we notice that the margin-based calibration doesn't work well in this setting. **(2)** The Temperature-based (T=5, 10) and the Angle-based calibrations can outperform the Logit loss by a large margin, especially the Angle-based calibration.
> >
> > Table 3: Targeted transfer success rate (%) when varying the target from the high-ranked class to low. (Average of 5 times)
> > |      | 2nd  | 10th | 100th | 200th | 500th | 800th | 1000th |
> > | :-:  | :-:  | :-:  | :-:   | :-:   | :-:   | :-:   |  :-:   |
> > |Logit | 83.7 | 83.2 | 77.3  | 74.5  | 71.5  | 64.9  |  52.4  |
> > |CE    | 77.4 | 58.6 | 34.0  | 26.9  | 23.7  | 16.7  |  7.0   |
> > |CE/5  | 91.3 | 88.7 | 80.7  | 77.1  | 75.8  | 70.1  |  58.8  |
> > |CE/10 | 89.0 | 87.8 | 82.8  | 81.0  | 79.2  | 73.5  |  62.5  |
> > |Margin| 87.4 | 81.7 | 67.4  | 61.3  | 51.6  | 43.1  |  23.0  |
> > |Angle | 92.4 | 89.1 | 82.2  | 80.3  | 79.2  | 76.1  |  66.3  |

---

> > > ### Comment · Reviewer_WQZv · 2022-08-05
> > >
> > > Thanks for providing the response.  I am worried all of these changes are falling more into a major revision and would not be within the limit of the conference.

---

> > > > ### Author Response · Authors · 2022-08-07
> > > > **Summary of Revision**
> > > >
> > > > Thanks for your comments and constructive feedback. We have uploaded a revision to address the concerns. The notable changes are below.
> > > >
> > > > 1. We revised the introduction section to clarify the distinction between the Logit [30] and moved the logit margin figure (Fig. 1) to the introduction to better illustrate the motivation and contribution of this study (Section 1).
> > > >
> > > > 2. We thoroughly rewrote the related work section to address the similarity issue pointed out by the reviewer qX4X (Section 2).
> > > >
> > > > 3. We added the experiments on the varied targets and T=10/20 in the combining logit calibrations. Besides, the suggestion for achieving a better-targeted attack is added (Section 4).
> > > >
> > > > 4. To better present the tables, we reported the average targeted transfer success rate with three digits at most for Tables 1, 2, & 3 instead of the average number of successfully attacked samples with four digits (Section 4).
> > > >
> > > > 5. We carefully polished the manuscript and corrected some typos and grammar mistakes.
> > > >
> > > > 6. More experiment results during the rebuttal period are added into the supplementary.

---

### Official Review · Reviewer_Boef · 2022-07-09

**Rating:** 5
**Confidence:** 4
**Soundness:** 3 good
**Presentation:** 2 fair
**Contribution:** 2 fair

**Summary:**

This paper designs a new logit calibration method which is inspired by knowledge distillation. The method uses logit calibrations in the CE loss function so that it can improve the targeted adversarial attack with higher transferability than other attack methods with cross-entropy loss. Except for the primary temperature-based method, this paper designs margin-based and angle-based methods to solve different surrogate models and different norms.

**Questions:**

Firstly, for the influence of different $T$ in CE, this paper claims that when the surrogate model is VGG16 and Inc-V3, a larger $T$ obtains better transferability. However, I am curious about is there a limitation of $T$ on VGG16 and IncV3. For example, after the ASR achieves 600, the performance of the targeted attack will decrease when $T$ continually increase. And then, although this method is based on CE, it would be better if the authors designed a new name to describe it.

**Limitations:**

The relationship between temperature-based, margin-based, and angle-based logit calibration is unclear. This paper claims that the margin-based one is designed to face different surrogate models, and the angle-based one is designed to solve the influence of various norms. However, in the experiments, the performance on T=5, T=10, Margin, and Angle does not prove the relation between them. This paper does not evaluate which method is the best for the targeted attack, or in other words, which option should I choose if I need to achieve the highest attack success rate?

**Strengths And Weaknesses:**

The strengths of this paper are:

This paper designs a new cross-entropy (CE) loss function to improve the targeted adversarial attack, which performs better than Logit (NIPS21).

Except for the temperature-based method, this paper designs margin-based and angle-based methods to solve different surrogate models and norms.

The weakness of this paper are:

This paper follows `Zhengyu Zhao, Zhuoran Liu, and Martha Larson. On success and simplicity: A second look at transferable targeted attacks. NeurIPS, 34, 2021.' from academic writing skills and code in specific. However, instead of Zhao et al. designing the Logit loss and using it to generate universal adversarial perturbations, this paper's method does not have any additional functions such as UAP.

Moreover, this paper's method only exceeds the Logit loss by around 10%, which is not a significant improvement. Therefore, this paper lacks novelty.

The equation 14 seems to have some mistakes. On the left of the equation, would $z_{i}$ be $\tilde{z}_{i}$?

A few grammar problems in this paper should be improved. For example, in line 275, it should be "be similar to"; in line 23, it should be "Following many approaches"; in line 27, it should be "it is vital to explore."

---

> ### Author Response · Authors · 2022-08-01
> **Response to Reviewer Boef**
>
> Thanks for your comments. We would like to address your concerns in the following aspects.
>
> >**Comment 1:** The influence of different $T$ in CE.
> >
> >**Response 1:**
> >
> >(1) ***A large $T$ for VGG-16 and Inc-V3:*** In the Response to Reviewer SQcN, we guess that the T is related to the model depth, in which a large $T$ is preferred for the CNN models with few layers. Compared with the ResNet-50 and DenseNet-121, VGG-16 and Inc-V3 have few layers, and better performance is obtained using large $T$.
> >
> >(2) ***The results of continually increasing $T$:*** In the supplementary, we analyzed the relation between the Logit loss in [30] and the CE calibrated by a large $T$. The gradient of Logit loss is $\frac{\partial L_{Logit}}{\partial \phi(\hat{x})} = - W_t$, and the gradient of CE with a large $T$ is $\frac{\partial L_{CE}^T}{\partial \phi(\hat{x})} \approx - \frac{W_t}{T}$. Since the I-FGSM only considers the Sign of gradient while neglecting the magnitude, then the optimization of the CE calibrated by a large $T$ is nearly equivalent to the Logit loss. In Table 1 and Figure 1 in the supplementary, we reported the comparison results of $T=50, 100$, and the Logit loss. Their results are very similar to each other, which verifies our analysis between the Logit loss and using a large $T$ in CE.
> >
> >Therefore, the performance of the targeted attack will get saturated when T continually increases since it is nearly equivalent to the Logit loss.
>
> >**Comment 2:** The relation between three calibration methods.
> >
> >**Response 2:** In this study, we investigate temperature-based, margin-based, and angle-based logit calibrations to validate the main hypothesis of our study that “enlarging the logit margins can increase the targeted transferability.”
> >
> >The temperature-based is the simplest one which only calibrates the logits by a constant value of T.  However, the optimal T is different for different models, as shown in Tables 1 & 2. Therefore, we investigate the margin-based and angle-based calibrations to deal with this hyper-parameter issue.  The margin-based method adaptively computed the “T” based on the Top-2 logits of each iteration instead of using a constant value. On the aspect, since $z_i= W_i*x + b$ and the L2 norm of $W_i$ is different for each class $i$, we further perform the calibration by normalizing the classifier weight $W_i$ of each class $i$ and the feature $x$ to the unit length by L2-normalization. This calibration is actually computing the cosine between the $W_i$ and $x$ while without considering their norms. Therefore, we term it angle-based calibration.
>
> >**Comment 3:** The option for the best targeted attack.
> >
> >**Response 3:** Since the best combinations of different models are different, we currently cannot have a universal receipt for each model. Based on the results in the manuscript and new results in the rebuttal, we might suggest using (T=5 + Margin) or (T=5 + Angle) for CNNs with more layers and the single Margin-based calibration for CNNs with fewer layers.
>
> >**Comment 4:** Typos and Grammar mistakes.
> >
> >**Response 4:** We will carefully polish the manuscript.

---

### Official Review · Reviewer_SQcN · 2022-07-11

**Rating:** 6
**Confidence:** 5
**Soundness:** 3 good
**Presentation:** 2 fair
**Contribution:** 3 good

**Summary:**

The authors propose a novel and effective method to improve the transferability of adversarial attacks. They increase the logit margins between targeted and non-targeted classes, which can quickly become saturated in cross-entropy loss.

**Questions:**

Could authors provide any interpretation of the results in Table 1? For example, why do Margin and Angle have better performance when the surrogate models are VGG17 and Inc-v3, but have lower performance for ResNet50 and Dense121?

**Strengths And Weaknesses:**

Strengths:
1. The findings are very interesting and the motivation is well-explained.
2. Comprehensive experiments are presented and the combining logit calibrations have significantly better performance than previous methods.

Weaknesses:
1. The proposed method has various settings and hyper-parameters. Compared to the simple Logit method, the proposed method needs more effort for tuning or need combining logit calibrations to achieve better performance. This can make the method less attractive to the community.
3. There is no theoretical analysis to support the empirical findings.
2. The presentation of the results needs to be improved. All tables contain tons of numbers, which makes it hard for the reader to get the point in a short time.

---

> ### Author Response · Authors · 2022-08-01
> **Response to Reviewer SQcN**
>
> Thanks for your comments. First, we will revise all tables to have a better presentation in the revision. Then, we would like to address your concerns about the interpretation of Table 1.
>
> >**Comment:** Potential interpretation of the results in Table 1.
>
> >**Response:** We argue the main reason for better results obtained by Margin-based calibration for the VGG-16 is mainly due to the influence of model depth. For the CNN models with fewer layers, a large normalization factor “T” is preferred to achieve higher targeted transferability. In our Margin-based calibration, the denominator “T” (logit margin between the first and second logits) will keep increasing along with the optimization iterations and thus leads to better performance.
> >
> >To further check the influence of depth, we leverage the ResNet-18 with fewer layers as the surrogate model and reported the results in the following Table 1. We also find that a large T and the margin-based calibration are preferred.
>
> >**Table 1.** The average number (#) of successfully attacked targeted samples with the ResNet-18 as the surrogate model.
> |       | Inc-v3         | ResNet-50       | Dense-121       | VGG-16 |
> | -     | :-:       | :-:        | :-:        |:-: |
> |CE     |21/30.4/29.6    |191.8/239.8/259.8|185.8/239.6/246.2|158.6/192.6/190.4|
> |CE/5   |39.2/108/119    |278.2/606.8/636.2|271.8/574.8/615.6|237.2/530/565.8|
> |CE/10  |36.2/111.6/132.4|259.2/597.4/668.6|258.6/571.8/642.2|222.4/530/596.6|
> |CE/20  |38.8/113.8/129.8|251.8/578/641.8  |248.2/543.2/607  |211.4/497.4/571.2|
> |Margin |41/113/130.8    |273/601.4/653.2  |273.2/572.6/629  |234.2/535.4/586.2|
> |Angle  |36.6/81.8/83.6  |271.4/514.8/542.6|280.6/527.6/557.4|239/449.4/462|
> |Logits |37.2/100.2/122  |247.8/556.2/606.8|243.2/536.4/585. |212.4/494.2/548.6|

---

### Official Review · Reviewer_qX4X · 2022-07-12

**Rating:** 2
**Confidence:** 5
**Soundness:** 2 fair
**Presentation:** 1 poor
**Contribution:** 2 fair

**Summary:**

This work proposed three different calibration methods, temprature-based, margin-based and angle-based temperature scaling to enlarge the margin between targeted logit and non-target logits to improve transferability of targeted adversarial attacks. This work is highly inspired by the work [1] and perform experiments to show the proposed methods are better than other existing methods.


[1] "On Success and Simplicity: A Second Look at Transferable Targeted Attacks".
Zhengyu Zhao, Zhuoran Liu, Martha Larson. NeurIPS 2021.

**Questions:**

In general, it leaves me a poor impression when I realize the great similarity between the Related Works in this work and that in [1]. Although the authors try to reframe the sentences, it's still very unprofessional to structure related works with such a strong similarity with another existing work.

**Limitations:**

See above.

**Strengths And Weaknesses:**

First of all, after comparing the related work in [1] and this work, there is a huge amount of overlapping of the equations or rewriting the sentences. This significantly destroys the overall quality of the work.

Second, the improvement of the proposed method over [1] is marginal compared to the improvement of [1] over cross-entropy loss.

Third, the contribution of this work is limited. Although the authors proposed different temperature-scaling based methods to improve transferability of targeted attacks, which only achieve limited experimental gains, this work did not provide extra useful insight to this research area.

[1] "On Success and Simplicity: A Second Look at Transferable Targeted Attacks". Zhengyu Zhao, Zhuoran Liu, Martha Larson. NeurIPS 2021.

---

> ### Author Response · Authors · 2022-08-01
> **Authors' Response**
>
> Thanks for your feedback. We would like to address your concerns in the following two aspects.
>
> >**Comment 1:** The related work section.
> >
> >**Response 1:** The structure of our current related work is mainly based on the following considerations. (1) The [1] highly inspired this study, and we followed the academic writing skills of [1] to some extent. (2) The I-FGSM, MI-FGSM, TI-FGSM, and DI-FGSM have been used as the baseline in the experiments. Besides, the optimization of only using the Sign of gradient in the I-FGSM is essential for our analysis of the relation between Temperature-calibration with large T and the Logits loss function. (3) The Po-Trip and the Logits are two main comparison methods, and then we also introduce them in detail.
> >
> >We will rewrite the related work section in the revision to avoid this similarity issue.
>
> ----
>
> >**Comment 2:** Marginal Improvement and Contribution.
> >
> >**Response 2:** We totally disagree with your comments that the contribution of this work is limited by only achieving marginal experimental gains.
> >
> >* First, we would like to **recap the primary goal of this study**, which mainly aims to analyze why the widely used CE loss function can not generate adversarial samples with higher targeted transferability. However, previous studies only reveal this issue due to the vanished gradient issue without further analysis. In this study, we take a close analysis of the CE loss and find that the logit margin between the targeted and non-targeted classes quickly gets saturated during the optimization process, hindering the CE's transferability.
> >
> >*  Second, **how to solve this issue**? Based on our analysis, we then explored three different logit calibration methods to deal with the saturated issue of logit margin. The experiment results valid our findings for the problem. Besides, in the supplementary, we further analyze that the Logit loss in [1] is nearly equivalent to the Temperature-calibration with large T.
> >
> >*  Third, **what is not the goal**? We do not intend to beat the state-of-the-art by a large margin. Although the logit calibrations slightly outperform the Logit for most cases, they can significantly increase the performance of the original CE. Besides, we also notice the results of combined logit calibrations in Table 3, which can outperform the Logit by more than 10% when using the ResNet50 and Dense121 as surrogate models. The additional experiment on the difficult transfer with varied targets suggested by the Reviewer WQZv can further show the effectiveness of logit calibration in the targeted attack.
> >
> >Based on the above explanation, we believe our investigation in this study can provide valuable insight for future researchers by using the logit calibration from both attack and defense.

---

> > ### Comment · Reviewer_qX4X · 2022-08-08
> > **Thanks for the authors' response.**
> >
> > I would like to thank the authors for their response and have checked the revised version. I agree with Reviewer WQZv that the change of the current version is falling into a major revision. In particular, I would like to highlight the high overlap between the previous submitted manuscript and the existing work. Therefore, I remain my previous rating and still vote for reject.

---

### Author Response · Authors · 2022-08-07
**Summary of Revision**

We thank the reviewers for their positive comments and constructive feedback. We have uploaded a revised manuscript based on the reviewers’ feedback and have highlighted changes from the original submission in blue. We summarize the notable changes below.

1. We revised the introduction section to clarify the distinction between the Logit [30] and moved the logit margin figure (Fig. 1) to the introduction to better illustrate the motivation and contribution of this study (Section 1).

2. We thoroughly rewrote the related work section to address the similarity issue pointed out by the reviewer qX4X (Section 2).

3. We added the experiments on the varied targets and T=10/20 in the combining logit calibrations. Besides, the suggestion for achieving a better-targeted attack is added (Section 4).

4. To better present the tables, we reported the average targeted transfer success rate with three digits at most for Tables 1, 2, & 3 instead of the average number of successfully attacked samples with four digits (Section 4).

5. We carefully polished the manuscript and corrected some typos and grammar mistakes.

6. More experiment results during the rebuttal period are added into the supplementary.

---

### Meta-Review · Area_Chair_PGox · 2022-08-25

**Recommendation:** Reject
**Confidence:** Certain

**Metareview:**

In this paper, the authors propose novel method to improve transferability of targeted adversarial attacks by enlarging the margin between targeted logit and non-target logits.  Experiments on ImageNet with different methods demonstrated the effectiveness of the method. However, as is pointed out by the reviewers that there exist high overlap between the paper and the existing works, which significantly hinders the novelties of the paper. The paper are expected to clarify the novelty and provide more comprehensive evaluations.




**Award:**

No

---

### Decision · Program_Chairs · 2022-09-14

Reject